# Retrieval of Ocean Surface Wind Speed Using Reflected BPSK/BOC Signals

**Hao-Yu Wang \***  **and Jyh-Ching Juang**

Department of Electrical Engineering, National Cheng Kung University, Tainan City 70101, Taiwan; juang@mail.ncku.edu.tw
* Correspondence: N28044020@gs.ncku.edu.tw

**Abstract:** The Global Navigation Satellite System (GNSS) has become a valuable resource as a remote sensing technique. In the past decade, the use of reflected GNSS signals for sensing the Earth, also known as GNSS reflectometry (GNSS-R), has grown rapidly. On the other hand, with the continuous development of GNSS, multi-frequency multi-modulation signals have been used to enhance not only positioning performance, but also remote sensing applications. It is known that for some constellations, navigation satellites broadcast signals employing BPSK (binary phase-shift keying) modulation and BOC (binary offset carrier) modulation at the same frequency band. This paper proposes a new GNSS-R measurement, called a composite delay-Doppler map (cDDM), by utilizing the received reflected GNSS signals with different modulation techniques for the purpose of retrieving wind speed. The GNSS-R receiver can receive BPSK and BOC signals simultaneously at the same frequency band (e.g., GPS III L1 C/A and L1C or QZSS L1 C/A and L1C) and process the signals to generate GNSS-R measurements. Exploration of the observable features extracted from the composite DDM and the wind speed retrieval algorithm are also provided. The simulation verifies the proposed method under a configuration that is specified for the orbital and instrument specification of the upcoming TRITON mission.

**Keywords:** GNSS-R; TRITON; wind speed; BPSK; BOC

## 1. Introduction

It has been shown the Global Navigation Satellite System reflectometry (GNSS-R) can be used to observe numerous geophysical parameters above the Earth's surface, including, but not limited to, soil moisture [1–3] and sea ice [4–6]. The capability of GNSS-R for deriving ocean surface wind speed has also been validated in many studies [7–9]. Among all these applications, inversion of sea wind is the most popular subject in GNSS-R and is also crucial to weather forecasting. The algorithms for retrieving wind speed have been developing for about 30 years since 1990. During the initial period, scientists were merely observing variations by processing reflected GPS signals under different sea states. After Zavorotny and Voronovich first proposed the GNSS-R theory in 2000 [10], researchers began to match measured 1-D delay waveforms (DW) or 2-D delay-Doppler maps (DDM) with their local simulated counterparts to estimate ocean wind speed. However, this matching method is time-consuming and requires prior information (i.e., rough wind speed and wind direction). Subsequently, many studies have proposed to extract observables from the DW or DDM and correlate these observables to nearly coincident measurements using other wind sensors, such as buoys. To date, retrieval of wind speed with an empirical model by relating the observables with the collocated wind speed is the most common practice. A summary of the GNSS-R principles and other applications can be found in [11].

The Cyclone GNSS (CYGNSS) mission is a space-based GNSS-R mission, which comprises eight micro-satellites to provide measurements of the sea surface wind field with good spatial and temporal resolution. The evolution of the CYGNSS wind speed retrieval algorithm is briefly reviewed in the following. In 2014, Clarizia et al. [12] developed a preliminary approach to retrieve wind speed using the data recorded by the precursor mission (United Kingdom-Disaster Monitoring Constellation, UK-DMC). They presented a minimum variance estimator to composite five wind speeds estimated from five different observables derived from a small data set of GNSS-R DDMs. Subsequently, Clarizia et al. [13] presented the baseline CYGNSS L2 wind speed retrieval algorithm, with specific characteristics of CYGNSS in-orbit measurements. Instead of using five observables as in the previous study, they used two observations (i.e., DDMA and LES) to develop a weighted wind speed estimator. After CYGNSS was launched on December 15, 2016, Ruf and Balasubramaniam [14] developed a wind speed algorithm using a tremendous amount of on-board measurements, along with two different ocean surface wind speed reference sources. This time, they built independent wind-GMF by considering two different sea states, a fully developed sea (FDS) version and a young sea/limited fetch (YSLF) version. The YSLF GMF is designated for measuring high wind speeds, especially those of tropical cyclones. The overall wind speed retrieval performance of the CYGNSS mission was reported in [15]. Afterward, Park and colleagues, affiliated with National Oceanic and Atmospheric Administration (NOAA), also proposed an improved wind retrieval method in 2019 [16].

Recently, a few studies have been conducted to define new observables for wind speed retrieval. In [17], Gao et al. described the normalized delay waveform (NDW) width as a new observable, along with elevation angle and flight height, to retrieve wind speed. Their wind speed retrieval model has two versions: one is a multiple regression model, and the other is a neural network model. However, both models can only provide the same performance level as the matching method (i.e., comparing the measured delay waveform with a simulated delay waveform). In a study by Wang and co-workers [18], two new observables were proposed based on the variations in the DW distribution. They built a ground-based GNSS-R system to retrieve wind speeds under a gentle wind scenario and a typhoon scenario using the proposed observables by receiving and processing Beidou Geostationary Earth Orbit (GEO) satellite signals. The results demonstrated that optimal wind speed retrieval performance can be obtained by fine-tuning the threshold and coherent time before calculating the proposed observables. On the other hand, Juang et al. proposed a model-based approach that the relationship between the reflected delay waveform and direct delay waveform can be identified as a channel response function [19]. The authors claimed that the proposed method is insensitive to the variation of the transmitter power, and the remote sensing parameters (e.g., ocean wind speed) can be retrieved from the coefficients in the channel model. In addition, the channel response that is established from binary phase-shift keying (BPSK)-modulated signals can be directly applied to binary offset carrier (BOC)-modulated signals. The flight test in their study showed that the proposed method is feasible and potential. However, data retrieval performance of the proposed method has not been verified.

The first GPS III satellite was successfully launched on 23 December 2018 and went into service on 13 January 2020, after a series of rigorous on-orbit operational tests. The remaining nine satellites (IIIA block series) continue to be deployed. It is expected that the last satellite will be launched in the second quarter of 2023. GPS III is constructed to be fully backward compatible with existing GPS systems but with new capabilities related to both military and civilian use, including longer SV lifer, improved accuracy, and improved availability. The GPS III SV will transmit L1 C/A, L1 P(Y), L2 P(Y), the modernized L1M, L2C, and L2M, and new L1C and L5 signals. Among all these signals, the L1C signal, which makes GPS III interoperable with other satellite navigation systems, is a new additional civilian signal. It is believed that the benefits of L1C (including improved accuracy, additional navigation messages, and advanced anti-jamming capability) can provide civilian users with better PVT services throughout the globe. In this regard, it is of interest whether the new L1C signal of the GPS III can be used to enhance GNSS-R performance. In this paper, we proposed a

potential solution, described in Section 2, to incorporate this signal in the processing of wind speed retrieval. For a detailed description of the GPS III, including performance requirements, signal and system design, and arrangement of the program, readers are referred to [20].

Thanks to the contribution of TechDemoSat-1 (TDS-1) and CYGNSS, many advances have been made in the GNSS-R. The available data from these missions not only verify the feasibility of GNSS-R but also promote many potential applications in remote sensing. The TRITON satellite, conducted by Taiwan's National Space Organization, will draw on the experience of those missions to carry on a space-based GNSS-R mission to gather ocean surface roughness and wind speed for the purpose of weather research and forecasting. The purpose of this paper is to propose a new GNSS-R measurement by using the signals transmitted under different modulations (i.e., BPSK and BOC) at the same frequency band and source for retrieving ocean wind speeds. The proposed method is based on the difference caused by the reflected signal under different modulations. A benefit of the proposed method is that it may ignore the instrument calibration, which is a crucial pre-processing of scientific data generation, but a challenging procedure during on-board processing. The corresponding wind GMF that is using the derived observable was also developed in this paper. In this paper, the simulation parameters for generating and validating the proposed method are specified under TRITON specifications, including orbital measurement geometry, the nadir antenna gain pattern, and receiver hardware characteristics.

## 2. Materials and Methods

In this study, a new GNSS-R measurement for wind speed retrieval is proposed that utilizes the signal characteristics of next-generation GNSS under TRITON configurations. This section describes the data simulation under TRITON specifications, including the satellite tracks and reflection tracks. The real weather analysis data, which was used as the ground truth data, is also discussed. An explanation of the software used to simulate GNSS-R measurement is included. The proposed GNSS-R measurement, extracted observables, as well as the wind speed retrieval algorithm, are also described in detail in this section.

### 2.1. Orbital and Instrumental Specifications of TRITON Mission

The TRITON (initially called FORMOSAT-7 Reflectometry, FS-7R) satellite is a part of the FORMOSAT 7 program developed for a technology demonstration mission by the Taiwanese space agency, National Space Organization (NSPO) and scheduled to be launched in late 2021. Unlike other satellites, which were developed for the purpose of conducting the GNSS radio occultation (GNSS-RO) mission, the TRITON is designed to carry on the GNSS-R experiment. The mission objective of the TRITON is to measure the roughness and wind speed over the ocean surface based on the domestically-developed GNSS-R payload. In this subsection, we introduce the TRITON satellite trajectory and its corresponding reflection events. For more details about the TRITON mission, including the payload design and preliminary test, the reader can refer to [19,21,22].

The TRITON satellite has a sun-synchronous mission orbit at an altitude of 550–650 km. The on-board payload of the TRITON satellite is a GNSS-R receiver tailored for receiving and processing reflected GNSS signals, including GPS, QZSS, and Galileo. The TRITON satellite is equipped with one zenith antenna and one nadir antenna. The zenith antenna is a high gain dual-frequency (L1 + L2) antenna, which is used to receive line-of-sight GNSS signals for GNSS receiver to provide position, velocity, and timing information. The nadir antenna is a left-handed circularly polarized (LHCP) antenna with a gain value of 14.5 dBi at the L1 frequency and 12.7 dBi at the L2 frequency. It should be noted that the RF front-end behind the nadir antenna will sample the incoming signal at $16.368 \times 10^6$ samples per second. All these parameters are essential for the following simulations and are summarized in Table 1.

**Table 1.** Orbital and payload specifications of the TRITON satellite

| Parameter | Value |
|---|---|
| Orbit | Sun-synchronous |
| Altitude | 550–650 km |
| Period | 96 min |
| Nadir antenna gain | 14.5 dBi |
| Sampling rate | 16.368 MHz |
| Frequency | 1575.42 MHz |

As mentioned, the TRITON is orbiting the Earth at an inclination angle greater than 24 degrees, and is capable of measuring four simultaneous reflections on the Earth's surface every second. In the following, four GPS satellites with the highest elevation angle are selected to calculate the specular reflection position. Figure 1a shows the 24-h TRITON spatial coverage of ground tracks and the corresponding reflection points. Figure 1b also shows the spatial coverage of reflection points over the ocean only.

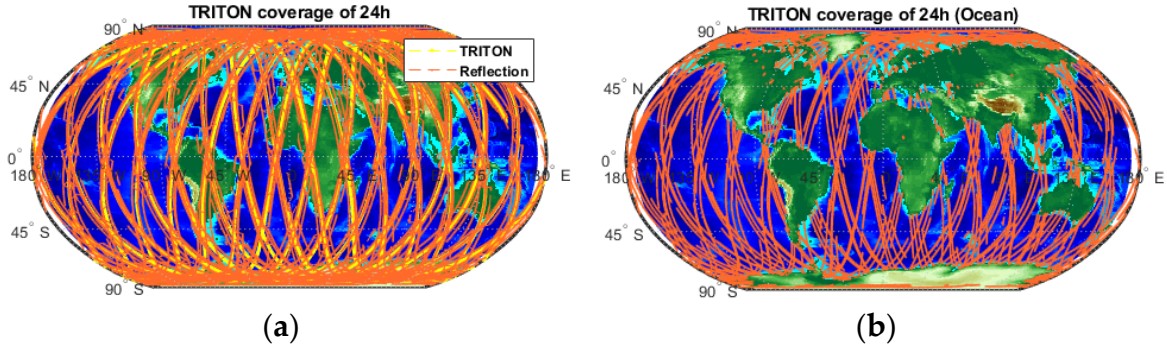

(**a**)　　　　　　　　　　　(**b**)

**Figure 1.** (**a**) Simulated TRITON ground tracks and reflection event coverage concerning GPS satellites for a 24-h period. (**b**) Shown is the reflection point over the ocean only. The broken yellow line indicates the ground tracks of the TRITON satellite, and the broken orange line indicates the reflection points from four GPS satellites with the highest elevation angle.

*2.2. The Ground Truth Data: European Centre for Medium-Range Weather Forecasts Product*

In the previous section, we simulated the trajectory of the TRITON satellite and its reflection tracks according to given parameters. However, it is also necessary to specify wind speed to generate the GNSS-R measurements, such as the delay-Doppler map (DDM) or the delay-waveform (DW). Therefore, we plan to employ the European Centre for Medium-Range Weather Forecasts (ECMWF) reanalysis product to generate the ground truth wind speed for the purpose of simulating the DDM and DW. To evaluate the retrieval performance of the wind speed of the proposed method, we used the ECMWF reanalysis product as 10-m-referenced ocean surface wind speeds. The reason to use real data, rather than a randomly generated wind speed value, is that the limit and distribution of wind speed values accord with the actual situation, which makes the simulation more realistic. ECMWF is a self-governing organization, and its core mission is to provide weather forecasts services and climate reanalysis products. The climate reanalysis product used here contains the wind speed information at the height of 10 m above the surface of the Earth with a spatial resolution of 0.25° × 0.25°, and temporal resolution of 1 h. Consequently, we used bicubic interpolation to estimate the reference wind speed at the locations and time according to the simulated reflection point of the TRITON satellite. In the following, we used the ECMWF data between 10 February 2020, and 16 February 2020, to produce interpolated 10-m-height ocean surface wind speeds as reference matchups. Figure 2 shows the observed wind speed distribution, which is obtained from the real ECMWF data by interpolation, according to given reflection tracks.

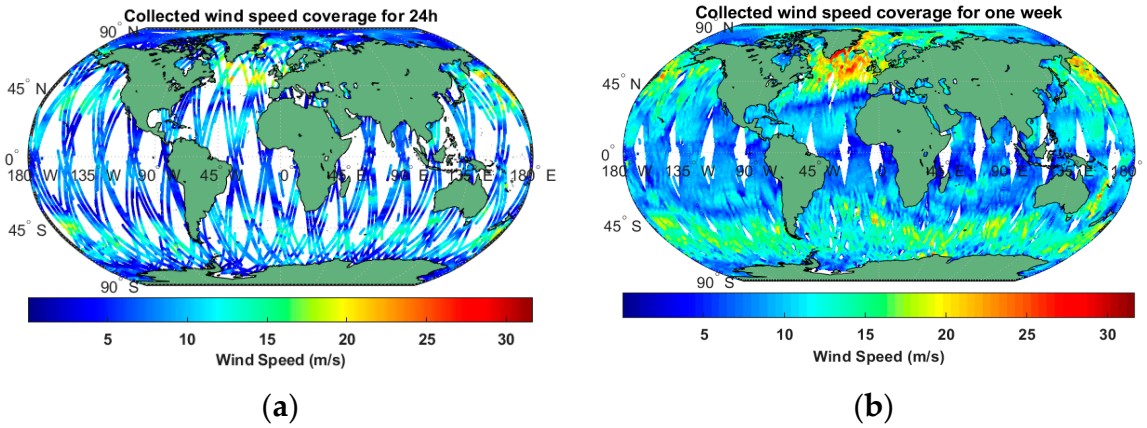

**Figure 2.** The TRITON reflection observation coverage, respectively, for (**a**) 24 h and (**b**) 7 days represented by wind speed. The color indicates the coincident wind speed obtained by interpolating the European Centre for Medium-Range Weather Forecasts (ECMWF) data.

*2.3. The Proposed GNSS-R Measurement: Composite Delay-Doppler Maps*

In the GNSS-R, one of the essential products is the delay-Doppler map (DDM). Usually, the DDM will be made in different sizes according to mission designs, such as $20 \times 128$ (Doppler bins $\times$ delay bins) for CYGNSS or $64 \times 128$ (Doppler bins $\times$ delay bins) for TRITON. In addition, the DDM resolution may also be different. In this paper, it is assumed that a GNSS-R receiver is capable of generating two types of DDMs. One is the DDM generated by processing the reflected L1 C/A signals (BPSK-DDM), and the other is the DDM generated by processing received reflected L1C signals (BOC-DDM). Under these conditions, the proposed GNSS-R measurement can be produced as derived below. As just mentioned, regardless of which size/resolution is specified, any number of pixel values in a DDM can be related to the input signal power as follows [23]:

$$C^M = G\left(P_a + P_r + P_s^M\right) \tag{1}$$

where $C$ represents the DDM values per delay-Doppler bin produced from the GNSS-R receiver in the unit of "counts"; $G$ is the total system gain; $P_a$ is the thermal noise power generated at the antenna, $P_r$ is the thermal noise power generated by the GNSS-R receiver, and $P_s^M$ is the received power of the scattered signal, with respect to L1 C/A ($M = BPSK$) or L1C ($M = BOC$). CYGNSS puts a great deal of effort into removing the effects of system gain and noise components from Equation (1), to retain a clear receiver scatter signal component via the black body calibration algorithm [24,25]. In this paper, we propose a new GNSS-R measurement utilizing the signal characteristics of L1 C/A and L1C without using any calibration methods. It is believed that both the BPSK-DDM and BOC-DDM are suffering from the same noise floor, because the received signals all go through the same instrument path. The DDM noise floor can be obtained by taking the average of a certain specific signal-free area in the DDM and can be expressed as:

$$C_N = G(P_a + P_r) \tag{2}$$

By removing Equation (2) from Equation (1), we obtain:

$$\overline{C}^M = GP_g^M \tag{3}$$

Finally, by taking the decibel computation and subtracting BOC-term from BPSK-term, we obtain

$$\Delta C_{dB} = 10 \cdot \log_{10}\left(P_s^{BPSK}\right) - 10 \cdot \log_{10}\left(P_s^{BOC}\right) \tag{4}$$

To implement the proposed method, we divided the procedure into three parts: (1) produce BPSK-DDM and BOC-DDM for each channel; (2) calculate the noise floor and eliminate it from the

DDM (the noise floor is calculated independently for each channel), and (3) apply the above equations to make the composite delay-Doppler map (cDDM). Figure 3 also shows the processing steps to generate the proposed GNSS measurement. An open-source GNSS-R simulator, WavPy, is used to simulate the proposed DDM measurement [26]. It should be noted that the WavPy currently employs the classic Z-V model [10] to implement the simulation.

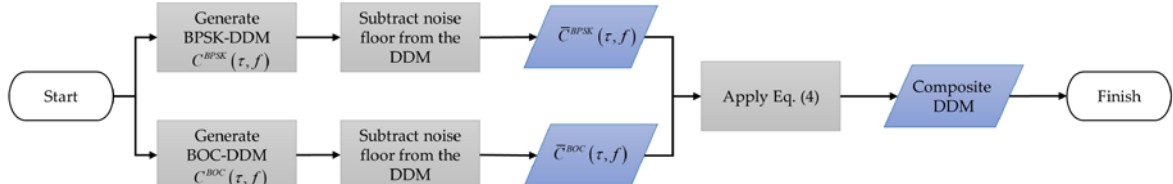

**Figure 3.** Flow chart of generating composite delay-Doppler map (DDM).

To analyze the relationship between the cDDM and the wind speed, we conducted a simple scenario with the parameters provided in Table 2. Figure 4 shows the simulated BPSK-DDM, BOC-DDM, and their corresponding CDDMs under different wind speeds and incidence angles. Then, we set a constant incidence angle of 30°, and simulate DW under different wind speeds, as shown in Figure 5b. In Figure 5a, we display an example of BPSK-DW and BOC-DW.

It is known that the autocorrelation function of the L1C spreading code has two side lobes, since the L1C spreading code employs the BOC modulation technique. The signal properties are changed due to the noise, fading, and distortion after reflection or scattering. This effect can lead to an extended trailing edge on the BPSK signal and asymmetrical side lobes on the BOC signal, as shown in Figure 5a. In Figure 5b, it can be seen that the area of the waveform around the peak varies with wind speed.

To further analyze the effectiveness of the cDDM in wind speed inversion, we simulate a large amount of data under various conditions, including different wind speeds and receiving geometry, based on realistic parameters. As mentioned in the preceding section, it is assumed that we collected the data generated from TRITON from 10 February 2020, to 16 February 2020. Therefore, a large set of cDDM is simulated according to real configurations, including antenna gain, incidence angle, and sampling rate. The actual ECMWF data during the selected period is used to generate the collocated reference wind speed. Additionally, we also apply some quality control to discard poor-quality simulated cDDM before further processing and analysis. First, measurements with receiving gains of less than 0 dB are excluded. Second, the BPSK-DDM and BOC-DDM, in which peak positions are too far from the theoretical center point, are excluded. Finally, the BPSK-DDM and BOC-DDM, in which the signal-to-noise ratio is less than 3 dB, are excluded. The remaining data set still counts up to 1,300,000 data pairs retained for analysis. We further split the data set into two independent sets: a randomly selected training set, which accounts for 80% of samples, and a remaining test set.

**Table 2.** Simulation parameters as a preliminary scenario.

| Parameter | Value |
| --- | --- |
| Incidence angle | 10, 20, 30, 40, 50, 60 deg. |
| Wind speed | 3, 5, 10, 15, 20, 30 m/s |

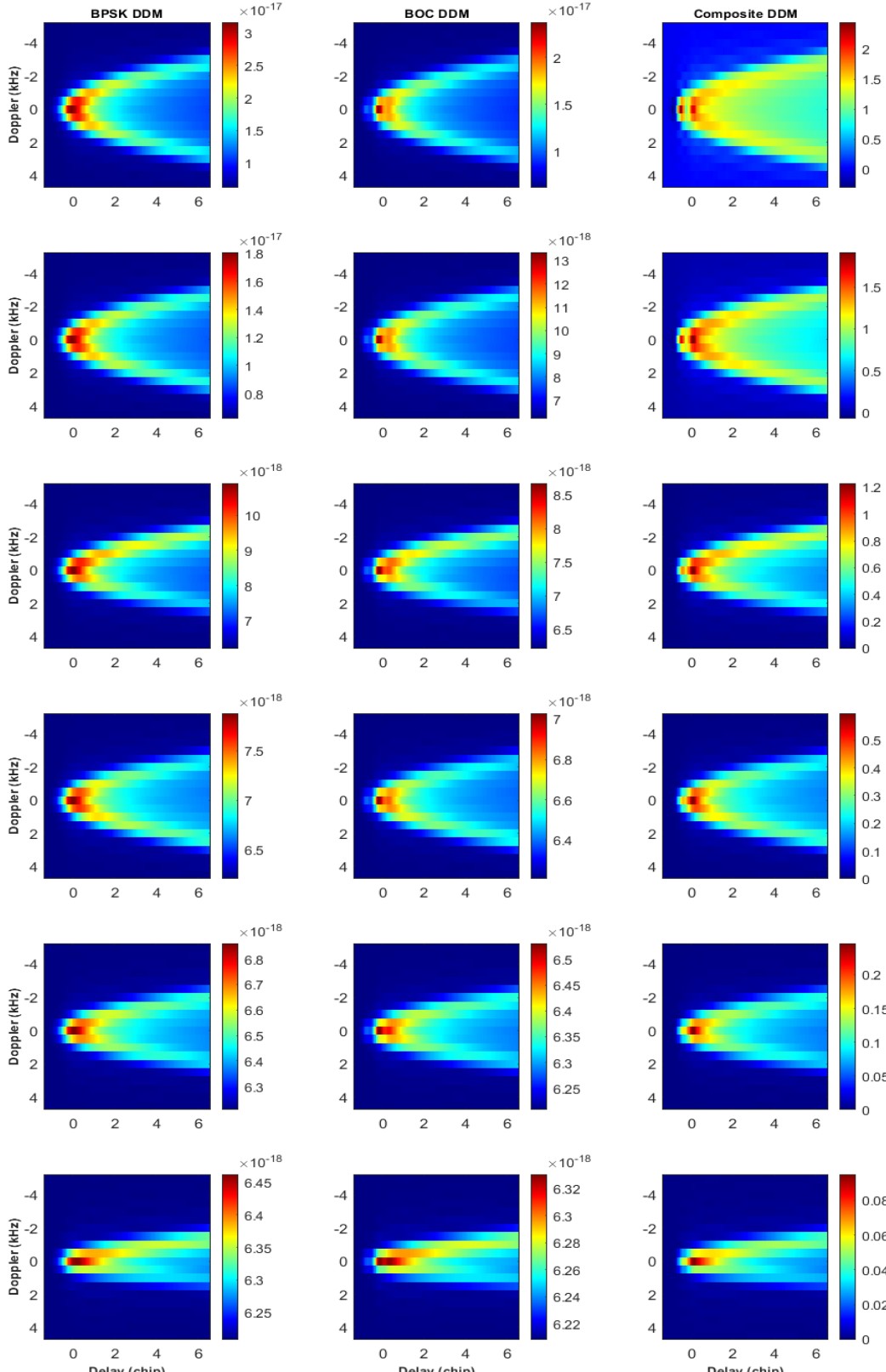

**Figure 4.** Examples of the binary phase-shift keying (BPSK)-DDM, binary offset carrier (BOC)-DDM, and the composite DDM (cDDM) for different wind speeds and incidence angles. The first, second, and third columns are BPSK-DDM, BOC-DDM, and cDDM, respectively. The specified parameters from the top row to bottom row are, respectively, 10/3, 20/5, 30/10, 40/15, 50/20, and 60/30 (incidence angle/wind speed).

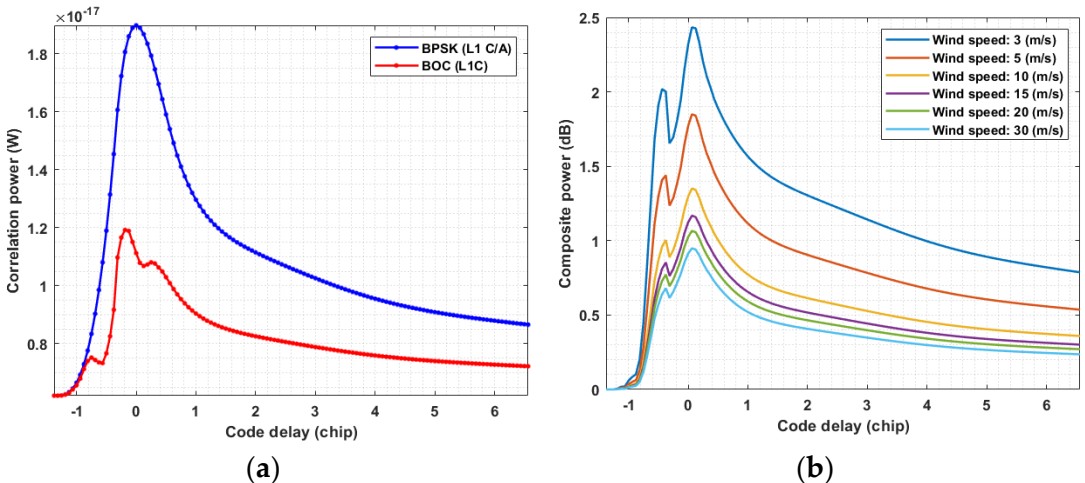

**Figure 5.** (**a**) Comparison of a BPSK-delay-waveform and a BOC-delay-waveform under a wind speed of 5 m/s. (**b**) Comparison of the composite delay waveform for different wind speeds at an incidence angle of 30°.

### 2.4. The Observable and Wind Speed Retrieval Algorithm

In principle, regardless of the type of GNSS-R measurement (i.e., delay-Doppler map or delay-waveform), one needs to extract useable observables from the GNSS-R measurement and regress the observable to the collocated wind speeds (or other geophysical parameters). Subsequently, a wind retrieval algorithm can be obtained by deriving a geophysical model function (GMF) that relates the observable to the wind speed, based on the regression parameters. In this study, we plan to extract three observables from the composite delay-Doppler map and deriving their corresponding GMFs. These observables are delay-Doppler map average, the delay-waveform average, and the delay-Doppler map peak, respectively. The following discussion provides a description of the observable and the procedure used to define the wind GMF.

The first observable is the so-called delay-Doppler map average (DDMA), which is a common observable and has been used in many studies on GNSS-R. The concept of the DDMA is the average signal power over a specified region of the DDM around the peak value position. The designer requirement determines the area in which to calculate the DDMA. For example, the Cyclone GNSS (CYGNSS) mission chooses the range of delay as (−0.25 0.25) chips and the range of Doppler as (−1000 1000) Hz, which corresponds to $3 \times 5$ bins of the DDM, to meet the mission requirement [13]. This selection can provide the retrieved wind speed with a spatial resolution of around 25 km $\times$ 25 km. In this paper, we employ the concept of DDMA on the cDDM to produce the composite delay-Doppler map average (cDDMA), which can be expressed as follows:

$$cDDMA = \frac{1}{MN} \sum_{m=1}^{M} \sum_{n=1}^{N} \overline{Y}_{cDDM}(\tau_m, f_n) \tag{5}$$

where $\overline{Y}_{cDDM}$ represents the value of the cDDM obtained from Equation (4); $M$ and $N$ are specified delay and Doppler range, respectively, and $\tau_m$ and $f_n$ are the delay and Doppler at indexes $m$ and $n$, respectively. We inherit the configurations from the CYGNSS; that is, a $3 \times 5$ cDDMA is calculated from the cDDM and used in the subsequent analysis. The second observable is the composite delay-waveform average (cDWA), which is the average value around a specified range in zero Doppler cDDM calculated using the following equation:

$$cDWA = \frac{1}{M} \sum_{m=1}^{M} \overline{Y}_{cDDM}(\tau_m, f_c) \tag{6}$$

where $f_c$ represents the Doppler index located at the zero Doppler of the cDDM. The last observable is the cDDM peak (cDP), which is defined as the highest value of the cDDM. It should be noted that all the above observables are in units of dB.

Before developing the GMF, one of the issues is the need to correct any effects that may affect the observable. Therefore, we first investigate the relationship between the observable and the nadir antenna gain at the specular point ($G_r$). Here, we take the DDMA observation as an example. Figure 6 shows the relationship between the cDDMA$_0$, derived from the simulated cDDM using real TRITON settings via WavPy, and the interpolated ECMWF wind speeds. Clearly, the observable has a dependence on $G_r$.

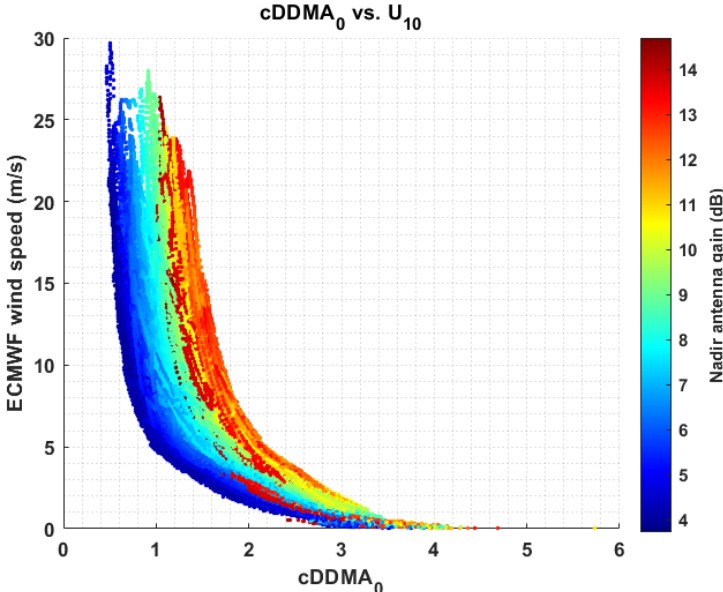

**Figure 6.** Relationship between composite delay-Doppler map average (cDDMA)$_0$ (dB) and ECMWF U10. The color indicates the nadir antenna gain value at the specular point.

To eliminate the effect of the antenna gain resulting from the observable, we apply the correction method used in [13]. The dependence of cDDMA on $G_r$ is shown in Figure 7a, where the cDDMA, corresponding to different values of $G_r$, takes the average of different wind speed values. Figure 7a shows a more clear dependence relationship between cDDMA and $G_r$ depicted in Figure 6. It is also shown that the dependence of the cDDMA on $G_r$ does not vary with wind speed, and we can, therefore, developed an empirical correction. Their value has normalized the data sets corresponding to a single wind speed value at $G_r = 0$ dB, and we find that a sum of sines function, given by $f(x) = a_1 sin(b_1 x + c_1) + a_2 sin(b_2 x + c_2)$, where $x = G_r$, can fit the normalized cDDMA data points in that curve. Figure 7b shows the normalized data points and their corresponding best fit polynomial function. The same procedure is applied to the other two observables, cDWA and cDP, since these two observables were also found to have a similar dependency relationship with $G_r$. Note that the above analyses were all conducted using the training data set. However, we found that the dependence on antenna gain is not entirely directly proportional. That is, there is a decrease in the observable decrease with increase in $G_r$ when $G_r$ is greater than a specific value. As a result, we inspected and observed the dependence of the BPSK-DDM-derived observable and BOC-DDM-derived observable, respectively, on $G_r$. The results showed that the dependence of the observables on $G_r$ is entirely directly proportional, but they differ from each other. Therefore, the composite combination described in the proceeding section causes the dependence phenomenon shown in Figure 7.

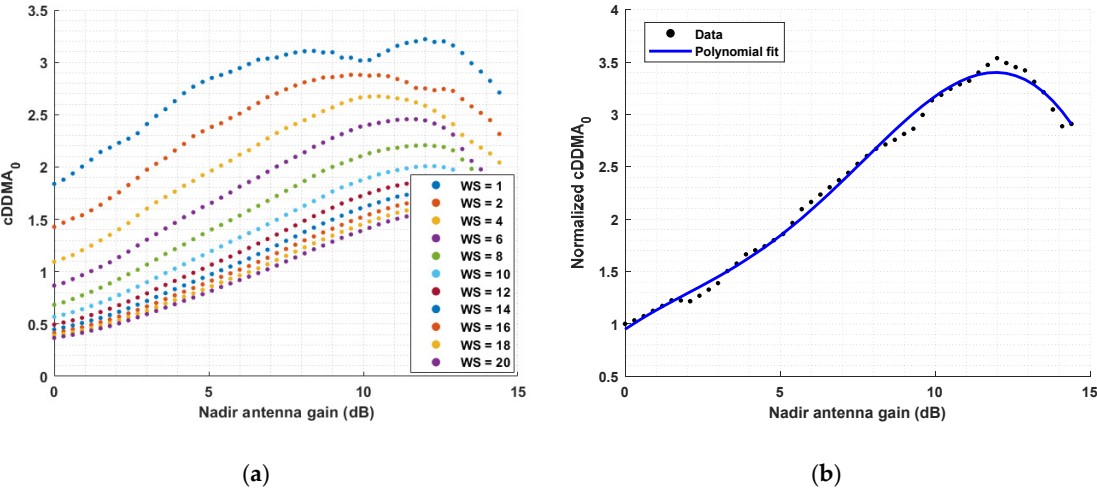

(**a**)                                     (**b**)

**Figure 7.** (**a**) cDDMA$_0$ versus nadir antenna gain at the specular point. The different colors indicate different wind speeds. (**b**) Normalized cDDMA$_0$ obtained from (**a**) for all gain values and wind speeds versus the nadir antenna gain, represented by the black dots. The blue curve is a fourth-order polynomial function that fits the data.

We can then obtain the corrected observable with the following expression:

$$O_1 = \frac{O_0}{f(G_r)} \tag{7}$$

where $O$ is the observable (i.e., cDDMA, cDWA, or cDP), and $f(G_r)$ is the polynomial fit as stated above. The cDDMA before and after the $G_r$ correction, represented as scatter density, is shown in Figure 8. The figure manifestly shows that the dependence on the nadir antenna gain value has been eliminated, as compared to Figure 6.

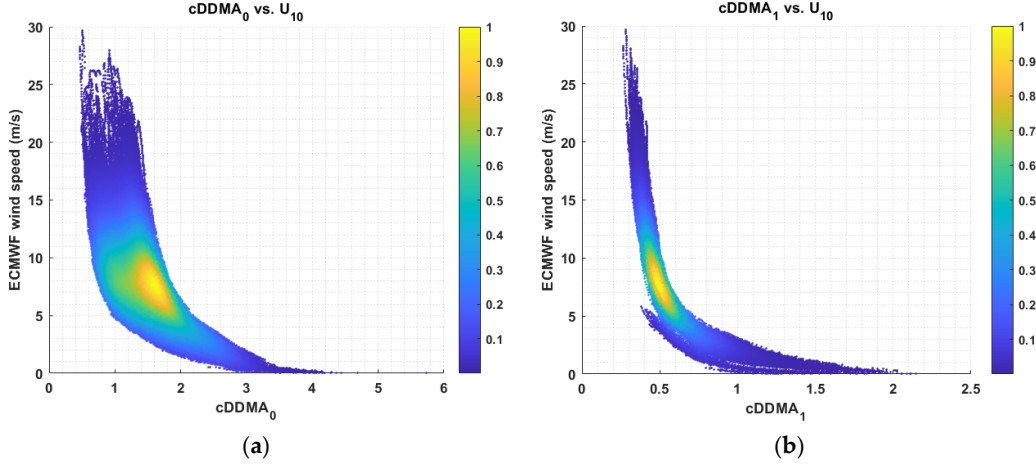

(**a**)                                     (**b**)

**Figure 8.** Scatter density plot of the cDDMA versus ECMWF wind speed before nadir gain correction (**a**) and after nadir gain correction (**b**).

The final observables used to develop the wind geophysical model function (GMF) are the corrected observables that are obtained from Equation (7). To develop the GMF, we regress the observable against the ECMWF wind speed. In this section, several commonly used fitting models were used to develop the GMF, including power model, exponential model, two-term exponential model, and polynomial model. In addition to the above equations, a dual-form model, which is found to best fit the observed and wind speed, was proposed for developing the GMF. Table 3 provides the mathematical expression for the models described above, which were also used for the subsequent processing.

**Table 3.** Comparison of fitting models. (In this table, $x$ is the observable, $U_{10}$ is the ECMWF wind speed, and the remaining symbols is the dependent parameters for each model).

| Model | Equation | |
|---|---|---|
| Power model | $U_{10} = a \times (x - c)^d + b$ | (T 3.1) |
| Exponential model | $U_{10} = b + ae^{-c(x-d)}$ | (T 3.2) |
| Two-term exponential model | $U_{10} = a_1 e^{b_1 x} + a_2 e^{b_2 x}$ | (T 3.3) |
| Polynomial model | $U_{10} = a_3 x^{-3} + a_2 x^{-2} + a_1 x^{-1} + a_0$ | (T 3.4) |
| Exp-power model (dual model) | $U_{10} = \begin{cases} a_l x^{b_l} + c_l, & x < x_{th} \\ a_h e^{b_h x} + c_h, & x \geq x_{th} \end{cases}$ | (T 3.5) |

Among these five models, the power model, the exponential model, and the two-term exponential model were performed non-linear least squares to find the fitting coefficient. The usage of these three models can be found in [9,27,28], respectively. In a study by Christopher and Rajeswari [14], the wind GMF was developed by dividing the observable into two parts, low wind and high wind, and then regressed each part with a polynomial form. Instead, we did not divide our data, since the behavior of the proposed observable is different from their research. We directly use a three-order polynomial form to regress the observable and the reference wind speed, as expressed in Equation (T 3.4). However, none of the above models, expressed in (T 3.1)–(T 3.4), can perfectly regress the behavior between the observable and the wind speed data. Therefore, we proposed a dual model, which can best fit the relationship between the observable and the wind speed data, to develop the wind GMF, as expressed in Equation (T 3.5). In the dual model, the data set was divided into two portions. The observable below a certain threshold (i.e., $O_{th}$) is used to determine $a_l$, $b_l$, and $c_l$ for the power form. The observable above a certain threshold (i.e., $O_{th}$) is used to determine $a_h$, $b_h$, and $c_h$ for the exponential form. The $O_{th}$ is determined when the least-squares residuals of the two portions are both minimized. Figure 9 shows the fitting results of using different models for training data. In the figure, the black scatter points represent the observable versus ECMWF wind speed, and the fitting curves are represented in different colors for each model. As shown in the figure, the dual model provides the best fit within the whole wind range, and the fitting curve of the exponential model has an obvious bias in low wind speed. However, if we try to reduce such a bias in low wind speed for the exponential model, the fitting result in high wind speed would become worse.

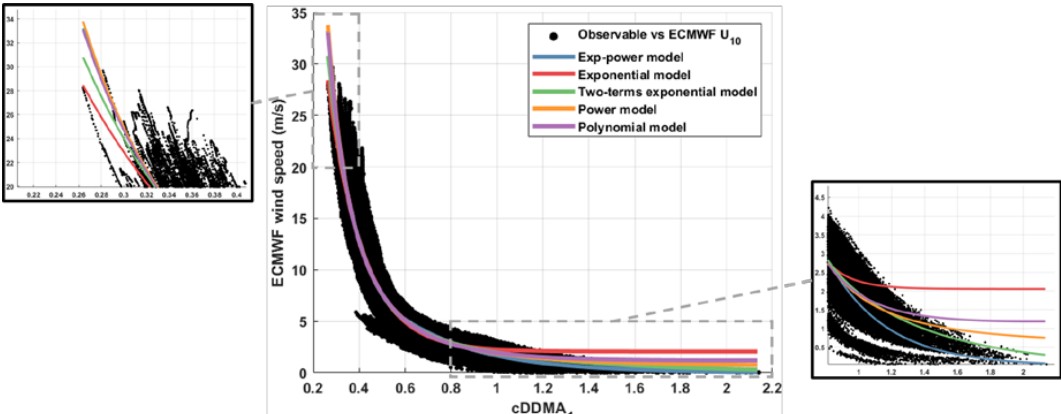

**Figure 9.** The composite delay-Doppler map average $cDDMA_1$ plotted against the ECMWF wind speed for the training data and the corresponding fit curve for different models.

To clearly compare the performance of wind retrieval between models, we calculate the mean errors and root-mean square (RMS) error using wind speed intervals of width ±1 m/s, as shown in Figure 10. It is shown that the retrieving accuracy and precision between different models are comparable to each other. However, the performance of the dual model is slightly better than other

models in low wind speeds, no matter in terms of mean error or RMS error. Thus, in Section 3, we conducted the performance evaluation using the dual model on different observables.

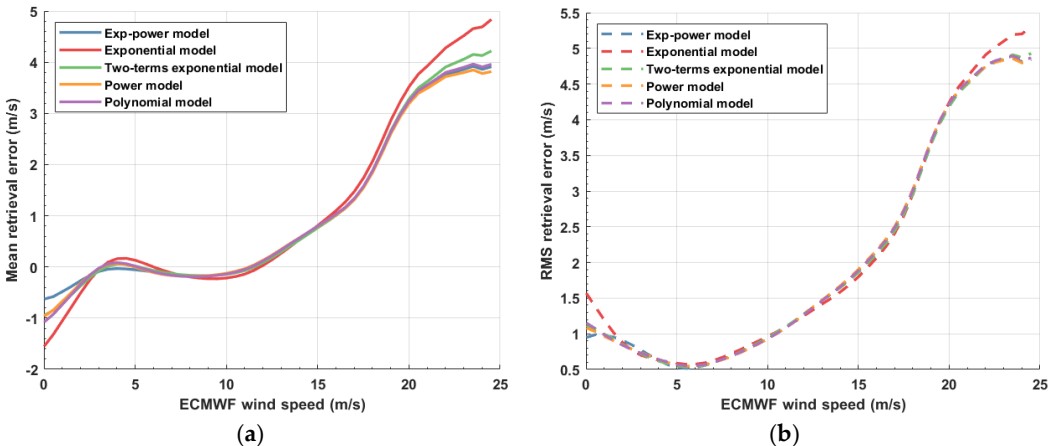

(a)　　　　　　　　　　　　　　　　　　(b)

**Figure 10.** (**a**) Mean error curves and (**b**) root-mean square (RMS) error curves for the retrieved wind speed using different models on test data.

### 2.5. Summary of the Proposed Method

The complete procedure used in this research can be divided into five parts, as shown in Figure 11. First, we employed the specific orbital parameters of the upcoming TRITON mission to simulate spacecraft ground tracks. Subsequently, the simulated ground tracks were used to calculate reflection points with respect to visible GPS satellite positions. Third, the ground truth for the wind speed data was generated by interpolating realistic ECMWF reanalysis data according to simulated reflection events. Fourth, the proposed GNSS-R measurements, the composite delay-Doppler maps (cDDMs), were generated via an open-source GNSS-R simulator, WavPy, using the observable derived from the previous step. Finally, we developed the independent wind GMF for the different observables. It should be noted that the purpose of this paper was to propose a new GNSS-R measurement that can be used to retrieve the ocean surface wind speed. The differences between the estimated wind speed from different observables and the ground truth wind speed were analyzed to evaluate the feasibility of the proposed method.

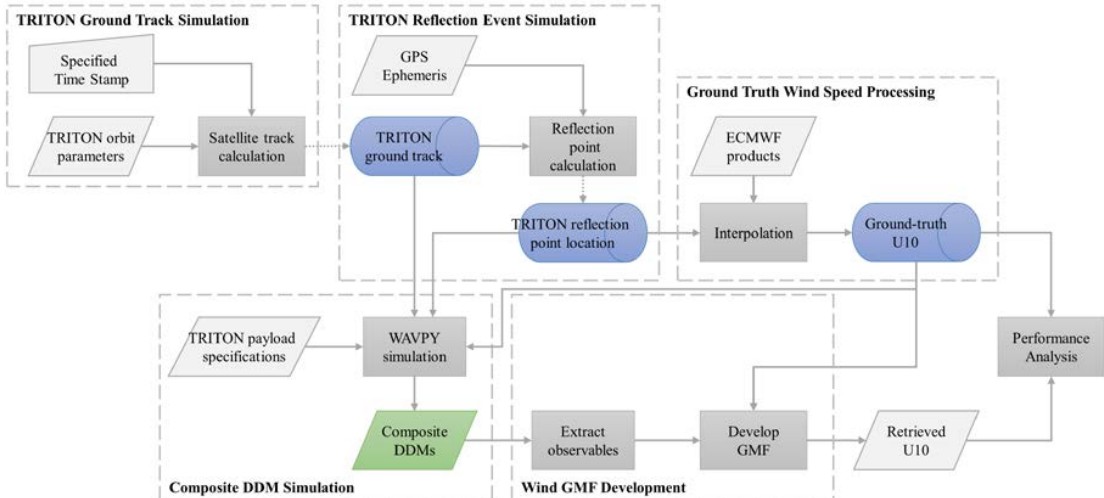

**Figure 11.** Overall research flow diagram, including the simulation of the TRITON ground track and corresponding reflection events, the proposed GNSS-R measurements and the observables, and the ground truth data generated for the performance evaluation.

## 3. Results

In this section, the performance of the retrieved wind speed using three observables from the proposed GNSS-R measurement is assessed using the test data set and the GMD derived from the dual model. Figure 12 shows the density scatter plot of the true wind speed and retrieved wind speed from different observables. As shown in the figure, the difference between the retrieved wind speed from the three observables is not noticeable. Nonetheless, we could still find that the use of $CDP_1$ caused less outlier branch behavior than when using $CDW_1$ and $CDDMA_1$, particularly with low to moderate wind speeds. Table 4 summarizes the overall statistical results, including the bias error, root-mean-square error, and the correlation coefficient of the retrieved wind speed using three observables. The statistical analysis indicated that the cDP-derived wind speed yielded the best performance, with an unbiased error, an RMSE less than 1 m/s, and a correlation coefficient of 0.96. This result is consistent with the data in Figure 12. However, the difference between wind speeds retrieved from the different observables is almost ignorable. The retrieved wind speeds all had an unbiased error, an RMSE value less than 1.1 m/s, and a correlation coefficient of more than 0.96.

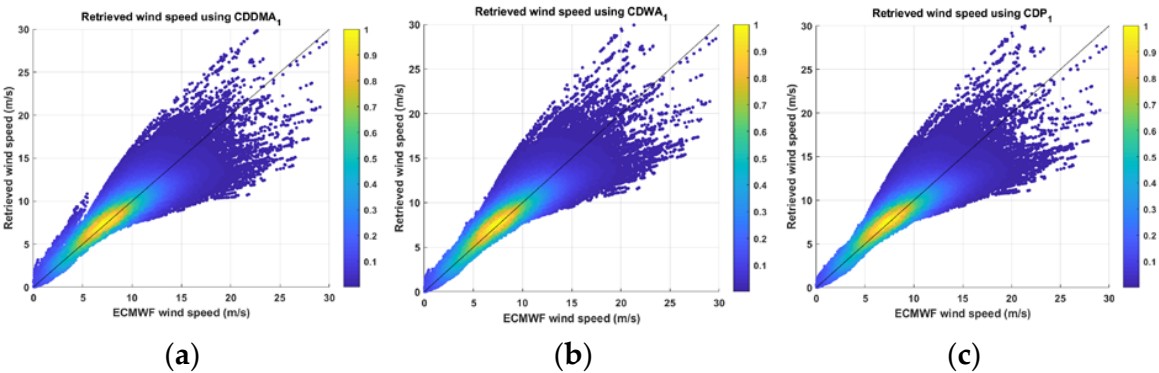

(**a**)             (**b**)             (**c**)

**Figure 12.** Scatter density plot of true wind speeds versus retrieved wind speed using (**a**) $cDDMA_1$, (**b**) $cDWA_1$, and (**c**) $cDP_1$, respectively, extracted from the composite delay-Doppler map test data. The solid black line represents the 1:1 agreement wind speed.

**Table 4.** Retrieved wind speed performance statistics for the different observables using the test data set. Bias and root-mean-square error (RMSE) are expressed in m/s. R represents the correlation coefficient between the ground truth wind speeds and the retrieved wind speeds.

| Observable | Bias | RMSE | R |
|------------|--------|--------|--------|
| cDDMA | −0.0057 | 1.0294 | 0.9603 |
| cDWA | 0.0002 | 0.9811 | 0.9640 |
| cDP | 0.0001 | 0.9657 | 0.9652 |

The obvious benefit from using cDP to retrieve wind speed is the higher geometric resolution, which is valuable when monitoring typhoon information. In this paper, however, we cannot affirm that using cDP with the proposed GNSS-R measurement is the best choice to retrieve wind speed, because the simulation is not able to reveal real ocean surface situations. Therefore, it is necessary to collect real data to test and evaluate the performance using different observables.

Additionally, the retrieval error under different wind speeds for all observables was also analyzed, as shown in Figure 13. The error plots were calculated using wind speed intervals of width ±1 m/s. It is clear that retrieving accuracy decreases as wind speed increases when the wind speed is more than 25 m/s. The figure also shows that the RMSE increases with increases in the wind speed when the wind speed is more than 15 m/s. Overall, the proposed method can provide unbiased retrieved wind speed with an RMSE of less than 2 m/s for wind speeds lower than 23 m/s. The reason for the worse retrieval accuracy and precision at high wind speeds is that the sensitivity of the observables on wind speed decreases, as shown in Figure 12.

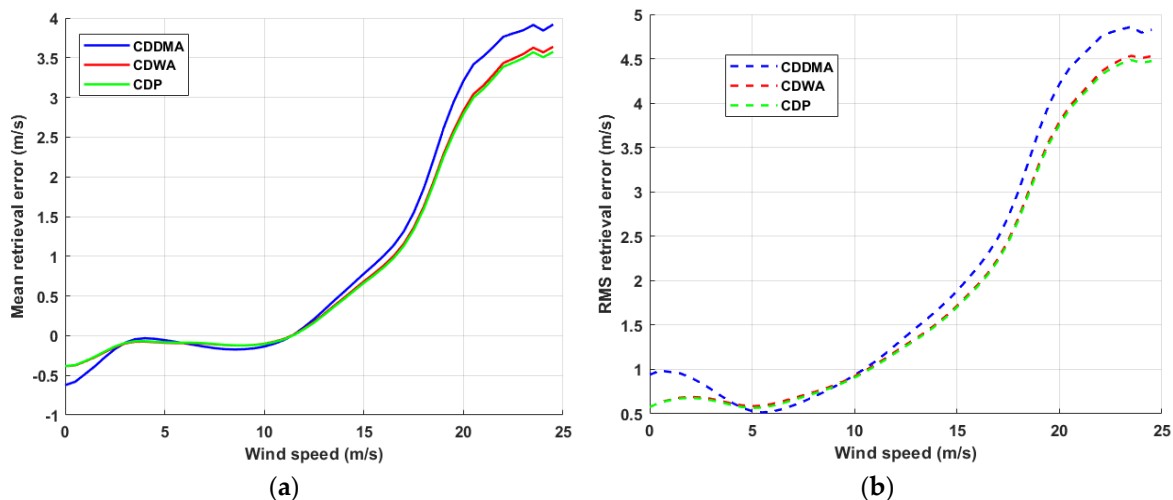

**Figure 13.** (**a**) Mean and (**b**) RMS retrieval error versus ground-truth wind speed for (blue) cDDMA, (red) composite delay-waveform average (cDWA), and (green) cDDM peak (cDP).

Figure 14 shows the probability density function (PDF) for the ground truth wind speeds and retrieved winds using different observables based on the proposed method. Obviously, regardless of which observables are used, the proposed method produces a PDF coincident with the reference PDF. In conclusion, the results obtained from the three observables have similar performance and yield impressive retrieval capabilities.

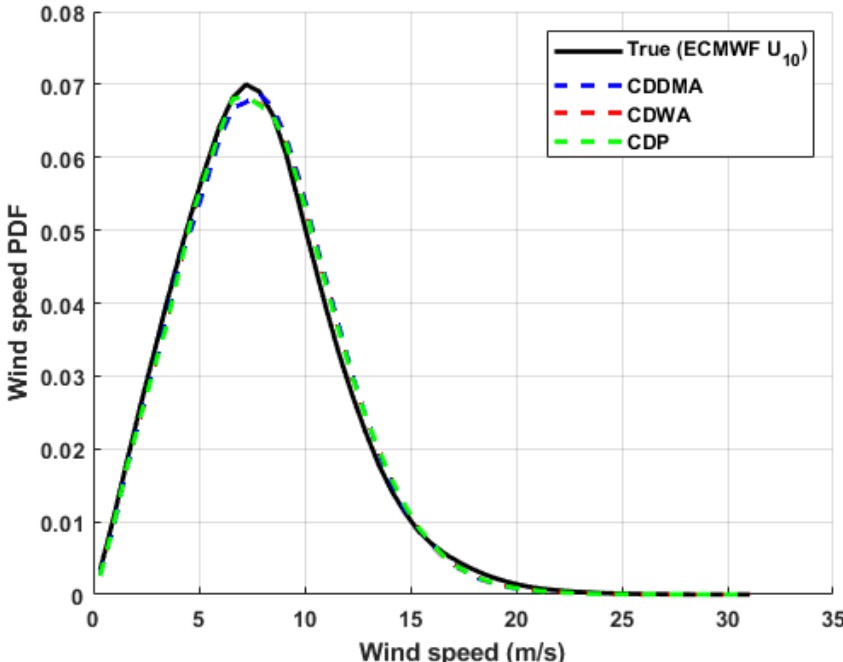

**Figure 14.** Probability density function (PDF) of reference winds (solid back) and retrieved winds using the proposed method with cDDMA (broken blue line), cDWA (broken red line), cDP (broken green line).

## 4. Discussion

This research describes a remote sensing method for deriving the ocean wind speed for the upcoming TRITON mission. On the other hand, as the United States proceeds to deploy GPS III satellites, the new signals are expected to benefit not only positioning services but also remote sensing performance. Therefore, we proposed a new GNSS-R measurement, composite delay-Doppler map (cDDM), by utilizing the properties in the new signals and used the proposed GNSS-R measurement

to develop a wind speed retrieval algorithm specifically for the TRITON. In addition, we also compared the performance of wind speed retrieval between different models. The comparison results indicated that the retrieval difference between different models is less. It could be concluded that the relationship between the observable and the geophysical parameter dominates the retrieval performance. Nonetheless, the proposed dual model yields better retrieval accuracy and precision than other models at low wind speed.

In addition, the performance evaluation of wind speed retrieval by using different observable was conducted. The results show that the observables extracted from the cDDM all could provide unbiased and high precision wind speed estimations. Furthermore, the derivation of the cDDM suggests that system calibration may be disregarded. This contribution was based on the hypothesis that different signal components (i.e., L1 C/A and L1C) suffer the same effects from noise and system gain. The algorithm has been tested with simulated DDMs that represent the expected features of the TRITON in-orbit measurements. However, many effects cannot be verified through simulations, such as ocean surface conditions around a tropical cyclone. It should be noted that the allocated transmission power for L1 C/A and L1C may vary among different GPS satellite vehicles. Future studies should account for the effect of L1 C/A to L1C transmission power differences between different navigation satellites when implementing the proposed method.

Compared with previous studies, as described in Section 1, this research is aimed at investigating the potential benefit of new generation GNSS signal characteristics for GNSS-R remote sensing. A new GNSS-R measurement, composite DDM, is proposed by processing and integrating BPSK-modulated signal and BOC-modulated signal. Through the derivation and simulation analysis, it seems that the proposed method is insensitive to the variation of the transmitter power and antenna gain, the factors which are of great concern in most of the conventional methods. The present study also compared the retrieval performance by using different fitting functions to develop GMF. In comparison, most of the previous work only develop their GMF using a specific fitting function. The reason may be limited by the relationship between the derived observable based on the conventional GNSS-R measurement and the remote sensing parameter. In this regard, only a slight difference is shown by modeling the relationship between observables that are extracted from the cDDM and wind speed using different fitting functions.

## 5. Conclusions

In this study, we propose and evaluate the effectiveness of a new GNSS-R measurement, the composite delay-Doppler map (cDDM), from the perspective of retrieving sea wind speed, by utilizing the signal characteristics of the next-generation GPS through a simulation. The ground track for the upcoming TRITON satellite, which is designed to perform GNSS-R mission, is simulated. The ECMWF data are used to provide the ground truth 10-m reference wind speed so that the DDMs including the proposed cDDMs which are regarded as the main observables in a GNSS-R mission can be generated. The cDDM is a combination of the DDM from BPSK signals and the DDM from BOC-modulated signals. As modern GNSS satellites broadcast navigation signals under these two modulations, the cDDM can be constructed. The paper emphasizes on the benefits of using cDDM for data retrieval. To this end, it is shown that the cDDM based method is less sensitive to the variations of the transmitted power and antenna gain. Three observations, namely, cDDMA, cDWA, and cDP, are extracted from the cDDM and associated with wind speed through a dual-model geophysical model function (GMF). The statistical analysis shows that the retrieved wind speed using cDP exhibits the best performance in comparison with cDDMA and cDWA. The paper provides a new and potential GNSS-R observable and processing approach on GNSS-R remote sensing. In the future, the cDDM based retrieval methods will be further investigated and its applications to TRITON will be studied.

**Author Contributions:** Conceptualization, H.-Y.W.; Formal analysis, H.-Y.W.; Methodology, H.-Y.W.; Project administration, J.-C.J.; Supervision, J.-C.J.; Writing—original draft, H.-Y.W.; Writing—Creview & editing, J.-C.J. All authors have read and agreed to the published version of the manuscript.

**Funding:** This research was funded by the National Space Organization, Taiwan, grant number NSPO-S-107151.

**Acknowledgments:** The authors would like to thank ECMWF for sharing the reanalysis data for the public. The authors would like to thank the reviewers for their reviews and comments that have helped to improve this article.

**Conflicts of Interest:** The authors declare no conflict of interest.

## Abbreviations

The following abbreviations are used in this manuscript:

| | |
|---|---|
| BOC | Binary Offset Carrier |
| BPSK | Binary Phase Shift Keying |
| cDDM | Composite Delay Doppler Map |
| cDDMA | Composite Delay Doppler Map Average |
| cDWA | Composite Delay Waveform Average |
| cDP | Composite Delay Doppler Map Peak |
| CYGNSS | Cyclone Global Navigation Satellite System |
| DDM | Delay Doppler Map |
| DDMA | Delay Doppler Map Average |
| DW | Delay Waveform |
| ECMWF | European Centre for Medium-Range Weather Forecasts |
| FDS | Fully Developed Sea |
| GMF | Geophysical Model Function |
| GNSS | Global Navigation Satellite System |
| GPS | Global Positioning System |
| LES | Leading Edge Slope |
| LHCP | Left Hand Circular Polarized |
| NDW | Normalized Delay Waveform |
| NOAA | National Oceanic and Atmospheric Administration |
| NSPO | National Space Organization |
| PVT | Position, Velocity, Time |
| QZSS | Quasi-Zenith Satellite System |
| RO | Radio Occultation |
| TDS | TechDemoSate |
| $U_{10}$ | Ocean Surface Wind Speed reference to a 10 m Height |
| YSLF | Young Sea/Limited Fetch |

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
