# Peer review of "Retrieval of Ocean Surface Wind Speed Using Reflected BPSK/BOC Signals"

_remotesensing, doi:10.3390/rs12172698_

Round 1
Reviewer 1 Report
the study is interesting; it investigates the retrieval wind through GNSS reflection. however, there are some points that should be improved before publishing:
- Major comments:
- the authors published related paper "https://ieeexplore.ieee.org/document/9082897/authors#authors" should be included in their study; while it's related to their application in this study.
- many parts can be summarized based on the author's previous studies; for instance, section 2.1 was included in "https://ieeexplore.ieee.org/document/9082897/authors#authors"
- the authors argued previous studies as presented inline 63; so I suggest make comparative results between this study and previous studies findings.
- the estimation of GNSS-derived wind vector at a selection region may improve the results section.
- minor comments:
- line 34: The algorithms for retrieving wind speed have been developing for about 20 years since 1990. it should be 30 years.
- the resolution of figures should be improved.
- the values scale "scale bar" of figure 8 should be presented.
Author Response
We are very much thankful for your valuable suggestions and comments.
Attached is our response.

Reviewer 2 Report
The subject of this paper is current and interesting ,the research is carried out rigorously and presents solid arguments for the conclusions made.Author Response
We would like to thank the Reviewer for taking the time and effort to review
our study. Thank you very much.

Reviewer 3 Report
Greetings,
Please find my comments below;
Major comments
1- You are using ECMWF data between Feb 10 and Feb 16, but tropical cyclone winds (16S) are not captured (winds over 70 knots) by your method, why?
2- What is the averaging time for your winds? Is it a 1-minute mean, or a 10-minute mean?
Minor comments
1- Please define all your acronyms.
2- Figure caption should be on the same page
3- Fig 2 use the same scale.
Line 31-32 “ to observe numerous geophysical parameters above the Earth’s surface, … and ocean surface wind speed “ surface winds are derived not observed.
Line 61-62 “An improved wind retrieval method was also proposed in [16] by the NOAA. “ Please rewrite this sentence.
Line 149-150 “Weather Forecasts (ECMWF) product to generate the ground truth wind speed for the purpose of simulating the DDM “ Please clarify. Are you using forecasts? It should be analyses.
Line 151-154 “The ground truth wind speed is also used to evaluate the wind speed performance that is retrieved using the proposed method. Additionally, using the ECMWF wind speed can make the simulation more realistic since the wind speed information includes the real wind strength and real distribution on the Earth’s surface. “ This is not clear, please rewrite. Also, it is incorrect to use the word truth. You are using an analysis.
Line 158 “ The ground-truth”, should be replaced. It is not ground-truth, it is just an analysis.
Thanks
Regards
Author Response
We are very much thankful for your valuable suggestions and comments.
Please see the attachment.

Round 2
Reviewer 1 Report
the authors have improved the manuscript; and the study can be published.